# Plasma Level of Circular RNA hsa_circ_0000190 Correlates with Tumor Progression and Poor Treatment Response in Advanced Lung Cancers

**DOI:** 10.3390/cancers12071740

**Published:** 2020-06-30

**Authors:** Yung-Hung Luo, Yi-Ping Yang, Chian-Shiu Chien, Aliaksandr A. Yarmishyn, Afeez Adekunle Ishola, Yueh Chien, Yuh-Min Chen, Tsai-Wang Huang, Kang-Yun Lee, Wen-Chien Huang, Ping-Hsing Tsai, Tzu-Wei Lin, Shih-Hwa Chiou, Chao-Yu Liu, Cheng-Chang Chang, Ming-Teh Chen, Mong-Lien Wang

**Affiliations:** 1Department of Chest Medicine, Taipei Veterans General Hospital, Taipei 112, Taiwan; hecterlo@gmail.com (Y.-H.L.); ymchen@vghtpe.gov.tw (Y.-M.C.); 2School of Medicine, National Yang-Ming University, Taipei 112, Taiwan; molly0103@gmail.com (Y.-P.Y.); chaoyuliu@hotmail.com (C.-Y.L.); mtchen@vghtpe.gov.tw (M.-T.C.); 3Institute of Clinical Medicine, National Yang-Ming University, Taipei 112, Taiwan; 4Department of Medical Research, Taipei Veterans General Hospital, Taipei 112, Taiwan; cschien6688@gmail.com (C.-S.C.); yarmishyn@gmail.com (A.A.Y.); aaishola01@gmail.com (A.A.I.); g39005005@gmail.com (Y.C.); figatsai@gmail.com (P.-H.T.); backyard0826@gmail.com (T.-W.L.); shchiou@vghtpe.gov.tw (S.-H.C.); 5School of Pharmaceutical Sciences, National Yang-Ming University, Taipei 112, Taiwan; 6Institute of Pharmacology, National Yang-Ming University, Taipei 112, Taiwan; 7Taiwan International Graduate Program in Molecular Medicine, National Yang-Ming University and Academia Sinica, Taipei 112, Taiwan; 8Division of Thoracic Surgery, Department of Surgery, Tri-Service General Hospital, National Defense Medical Center, Taipei 114, Taiwan; chi-wang@yahoo.com.tw; 9Taipei Cancer Center, Taipei Medical University, Taipei 110, Taiwan; leekangyun@tmu.edu.tw; 10Division of Pulmonary Medicine, Department of Internal Medicine, Shuang Ho Hospital, New Taipei City 235, Taiwan; 11Division of Pulmonary Medicine, Department of Internal Medicine, School of Medicine, College of Medicine, Taipei Medical University, Taipei 110, Taiwan; 12Division of Thoracic Surgery, Department of Surgery, MacKay Memorial Hospital, Taipei 104, Taiwan; wjhuang0@gmail.com; 13Department of Medicine, MacKay Medical College, Taipei 104, Taiwan; 14Division of Thoracic Surgery, Department of Surgery, Far-Eastern Memorial Hospital, New Taipei City 220, Taiwan; 15Department of Obstetrics and Gynecology, Tri-service General Hospital, National Defense Medical Center, Taipei 114, Taiwan; sundoor66@yahoo.com.tw; 16Department of Neurosurgery, Neurological institute, Taipei Veterans General Hospital, Taipei 112, Taiwan; 17Institute of Food Safety and Health Risk Assessment, National Yang Ming University, Taipei 112, Taiwan

**Keywords:** lung cancer, circular RNA, liquid biopsy, droplet digital PCR, programmed death-ligand 1

## Abstract

Lung cancer (LC) causes the majority of cancer-related deaths. Circular RNAs (circRNAs) were reported to play roles in cancers by targeting pro- and anti-oncogenic miRNAs. However, the mechanisms of circRNAs in LC progression and their prognostic value of treatment response remain unclear. By using next generation sequencing (NGS) of LC cell lines’ transcriptomes, we identified highly overexpressed hsa_circ_0000190 and hsa_circ_000164 as potential biomarkers. By using the highly sensitive RT-ddPCR method, these circRNAs were shown to be secreted by cell lines and were detected in human blood. Clinical validation by RT-ddPCR was carried out on 272 (231 LC patients and 41 controls) blood samples. Higher hsa_circ_0000190 levels were associated with larger tumor size (*p* < 0.0001), worse histological type of adenocarcinoma (*p* = 0.0028), later stage (*p* < 0.0001), more distant metastatic organs (*p* = 0.0039), extrathoracic metastasis (*p* = 0.0004), and poor survival (*p* = 0.047) and prognosis. Using liquid biopsy-based RT-ddPCR, we discovered the correlation between increased hsa_circ_0000190 plasma level (*p* < 0.0001) and higher programmed death-ligand 1 (PD-L1) level in tumor (*p* = 0.0283). Notably, long-term follow-up of the immunotherapy treated cases showed that upregulated plasma hsa_circ_0000190 level correlated with poor response to systemic therapy and immunotherapy (*p* = 0.0002, 0.0058, respectively). Secretory circRNAs are detectable in blood by LB-based RT-ddPCR and may serve as blood-based biomarkers to monitor disease progression and treatment efficacy.

## 1. Introduction

Lung cancer (LC) is the leading cause of cancer-related deaths throughout the world, and more than 85% of LC cases belong to non-small-cell lung carcinoma (NSCLC) [1]. Most LC patients have locally advanced disease or distant metastases at the time of diagnosis. In the past decade, the development of targeted therapies has led to significant improvement in the treatment efficacy for NSCLC, but the survival rate for advanced NSCLC still remains dismal [2]. Recently, immunotherapy for NSCLC has demonstrated significant survival benefits in certain patient populations [1,3,4,5,6,7,8]. Despite the marked advancement in the medical treatment of LC, fewer than 10% of advanced NSCLC patients survive for more than five years. Hence, the development of superior diagnostic and therapeutic approaches is critical for better understanding of the molecular pathogenesis of LC. In turn, such advancements can contribute to the discovery of novel biomarkers for cancer detection and molecular treatment targets for LC, as well as more personalized treatment of LC patients.

Circular RNAs (circRNAs) are non-coding RNAs that consist of a circular loop with multiple microRNA (miRNA) binding sites known as miRNA response elements (MREs); hence they function by competing with endogenous miRNAs to regulate gene expression [9,10]. Due to their closed structure, circRNAs are largely stable and very resistant to RNA-degrading endonucleases, therefore, they have much longer circulatory half-life as compared to linear RNAs, which qualifies them as promising disease biomarkers [10]. Many circRNAs exist in the cell cytoplasm and function in circRNA–miRNA–mRNA axes involved in proliferative and survival pathways in various types of cancer [11]. The prominent example of such axis is represented by ciRS-7 (CDR1as) circRNA that contains over 70 binding sites for miR-7, and functions as a miR-7 sponge [12]. The increased expression of ciRS-7 in tumor tissues leads to inhibition of miR-7 and subsequent activation of several oncogenes, including epidermal growth factor receptor (*EGFR*), rapidly accelerated fibrosarcoma 1 (*RAF1*), phosphoinositide 3-kinase catalytic subunit delta (*PIK3CD*), mammalian target of rapamycin (*mTOR*), and insulin receptor substrate-1 (*IRS-1*) [13,14,15]. Therefore, inhibition of circRNAs may block signaling pathways responsible for tumor growth and development [12,16,17,18,19]. However, the underlying biomolecular mechanisms and clinical correlation of circRNAs with lung carcinogenesis and disease progression are not yet well understood.

Tissue biopsies are used by clinicians for histological diagnosis and, more recently, for the genetic profiling of the tumor to predict tumor progression and response to treatment. However, the limitations of tissue biopsies such as procedure considerations, tumor heterogeneity, and dynamic genomic change of the tumor after treatment impede practicable repeated tissue biopsy assessment [20,21]. This hinders proper prospective and dynamic treatment and disease progression follow-up. On the contrary, liquid biopsy (LB), performed through a simple non-invasive blood test, can allow repeated and convenient analysis of the circulating tumor cues in the blood [10,22]. This enables consistent monitoring of disease progression, chemosensitivity, and treatment efficacy in a dynamic manner [23,24]. Unfortunately, the relatively low sensitivity and specificity of several LC-associated plasma biomarkers, including carcinoembryonic antigen (CEA), neuron-specific enolase (NSE), cytokeratin 19 (CYFRA-21-1), CA-125, and CA 19-9 makes it difficult to replace the existing X-ray-based imaging and tissue biopsies [24,25,26]. The droplet digital polymerase chain reaction (ddPCR) is a relatively newly developed nucleic acid detection method, which is based on the separation of PCR solution into individual droplets with independent reactions, thus ensuring single-molecule sensitivity.

Therefore, the main objective of this study was to identify potential circRNA biomarkers of LC, and apply the LB-based RT-ddPCR for cancer detection, diagnosis, and prediction of treatment efficacy in advanced LC. Next generation sequencing (NGS) was used to identify circRNAs overexpressed in LC cell lines with different EGFR mutation status, and two circRNAs, hsa_circ_0000190 and hsa_circ_0001649, were selected as potential biomarkers. These two circRNAs were shown to be secreted by LC cells in vitro, and were also detected in blood by RT-ddPCR. The expression of these two circRNAs was monitored in blood plasma of a cohort of LC patients, and was found to correlate with a number of pathological parameters and response to immunotherapy, thus confirming their usability as secreted biomarkers.

## 2. Results

### 2.1. Identification of Potential Secretory circRNA Biomarker Candidates Using RNA-Seq

Although increasing evidence has revealed that specific circRNAs, such as circRNA 100876 and hsa_circ_0013958, are upregulated in LC tissue comparing to adjacent normal tissue, little is known about the secreted circRNAs in patients’ peripheral blood [27,28]. Therefore, in this study, we aimed to identify such secreted circRNA biomarkers of LC. For this purpose, we designed a workflow, whereby RNA-Seq was first used to identify circRNAs most highly expressed in LC cell lines, followed by validation of their expression and secretion to the culture medium by qRT-PCR and RT-ddPCR (Figure 1A). RT-ddPCR was further used as a highly sensitive method to detect the levels of circRNA biomarker candidates in patients’ plasma, and their biomarker potential was assessed in a cohort of LC patients by identifying the correlation of their expression levels with different clinical parameters (Figure 1A).

To identify the potential circRNA candidates, we first performed next generation sequencing (NGS) of the transcriptomes of two distinct lung adenocarcinoma cell lines, HCC827 and A549, and a normal bronchial epithelial cell line BEAS-2B. The differential expression of candidate circRNAs was then examined among three distinct cell lines, including A549 (wild type EGFR lung adenocarcinoma cells), HCC827 (lung adenocarcinoma cells with an activating mutation in the EGFR tyrosine kinase domain), and the normal bronchial epithelial cells BEAS-2B according to the pipeline shown on Figure 1C. As was shown by principal component analysis (PCA), the circRNA transcriptomes of both LC cell lines were clearly more distinct from that of BEAS-2B (Figure 1B). In total, 7494 circRNAs were detected to be expressed in the analyzed cell lines with a cut-off >5 TPM (Figure 1C). These circRNAs were encoded by the genes distributed on all chromosomes (Appendix A) and the majority of the most differentially regulated circRNAs were cell line-specific (Appendix A). Among these 7494 detected circRNAs, there were 75 upregulated circRNAs (criteria: fold change > 5 between HCC827 and BEAS-2B and fold change > 1 between A549 and BEAS-2B) and 131 downregulated circRNAs (criteria: fold change < 0.2 between HCC827 and BEAS-2B and fold change < 1 between A549 and BEAS-2B) (Figure 1C). One of the criteria of the selection of potential biomarkers was the differential expression not only between cancer and normal BEAS-2B cell lines, but also between A549 and HCC827 LC cells different in mutation status of the *EGFR* gene, one of the major pathological parameters in LC. Therefore, we selected 65 upregulated circRNAs with higher expression in HCC827 than in A549 (HCC827 > A549) and 121 downregulated circRNAs with lower expression in HCC827 than in A549 (HCC827 < A549) (Figure 1C). By this narrowing down of the dataset, all upregulated circRNAs fitted the trend HCC827 > A549CC > BEAS-2B, and all downregulated circRNAs fitted the trend HCC827 < A549CC < BEAS-2B (Figure 1C). These selected circRNAs were further subjected to hierarchical clustering, and eventually, nine circRNAs with the most prominent fold changes were identified—among them four were upregulated and five downregulated (Figure 1C,D). These nine circRNAs included hsa_circ_0001649, hsa_circ_0000190, hsa_circ_0000325, hsa_circ_0058677, hsa_circ_0036574, hsa_circ_0036494, and hsa_circ_0014264, and hsa_circ_0038226, and hsa_circ_0026126 (Figure 1E). Collectively, our RNA-Seq analysis indicated that, compared to that in normal bronchial epithelial cells, hsa_circ_0000190 and hsa_circ_0001649 were highly upregulated in the lung cancer cell line A549 with wild-type EGFR expression, and were expressed at even higher levels in the lung cancer cells with constitutively active mutant EGFR. These two circRNAs were therefore selected as candidates and subjected to subsequent experiments for evaluating their potential to be liquid biopsy biomarkers of LC.

### 2.2. Validation of Expression of hsa_circ_0000190 and hsa_circ_0001649 in LC Cell Lines

To further explore whether hsa_circ_0000190 and hsa_circ_0001649 could be utilized for investigation of their potential as candidates for secretory circRNA biomarkers, we validated their cellular expression in LC cell lines by PCR methods. For this purpose, the divergent primers were designed that could specifically amplify the circRNAs across the backspliced junctions, but could not amplify conventionally spliced linear mRNA counterparts (Figure 2A). Indeed, as was shown by RT-PCR, the amplicons of the expected size (145 bp for hsa_circ_0000190 and 124 bp for hsa_circ_0001649) could be amplified from the cDNA of BEAS-2B, A549 and HCC827 cell lines, but not from the gDNA templates (Figure 2B). We also subjected these amplicons to Sanger sequencing and showed a perfect match of the sequencing results to the backspliced junction sequences from the circBase database (http://www.circbase.org/) (Figure 2C). As was further shown by qRT-PCR analysis, consistently with the NGS results, the expression levels of hsa_circ_0000190 and hsa_circ_0001649 were significantly upregulated in A549 and HCC827 cell lines as compared to BEAS-2B cells, with higher levels in HCC827 EGFR-mutant cell line (Figure 2D). In addition, to further eliminate the possibility that the PCR primers detect linear forms of RNA, we treated the total RNA with RNase R, which digests only linear form, but not circular form (Figure 2E). Compared with non-treated controls, the *GAPDH*-normalized relative levels of both circRNAs were significantly increased in RNase R-treated A549 samples due to the digestion of *GAPDH* mRNA and circRNAs remaining intact (Figure 2E). To summarize, these data indicated that the primers we used specifically detected the circular forms of hsa_circ_0000190 and hsa_circ_0001649, but not their linear counterparts, and that both circRNAs were highly expressed in LC cell lines but not the normal lung cells.

### 2.3. RT-ddPCR Detection of hsa_circ_0000190 and hsa_circ_0001649 Secreted by LC Cell Lines and in Human Blood Plasma

Accumulating data have revealed the advantages of using ddPCR in detecting cell-free RNA and DNA in a variety of human body fluids [29]. Indeed, we failed to detect hsa_circ_0000190 and hsa_circ_0001649 expression in human blood plasma by using conventional qRT-PCR (data not shown). To verify if hsa_circ_0000190 and hsa_circ_0001649 can be secreted from LC cells, we first collected the conditioned media from LC cell lines CL1-0 and CL1-5 and extracted the total secreted RNA for RT-ddPCR analysis (Figure 3A,B). RT-ddPCR was performed with optimized primer concentration and expression of both circRNAs was clearly detected by absolute quantification, whereby positive droplets were clearly distinguishable from the negative droplet background (Figure 3A,B). Furthermore, the expression of hsa_circ_0000190 and hsa_circ_0001649 was also detected in the conditioned medium of other LC cell lines to various extent, including HCC827 and A549 (Figure 3C,D). Agarose gel electrophoresis showed the anticipated size of the amplicons for both circRNAs in cell lysates of all tested cell lines (Figure 3E). These data indicate that hsa_circ_0000190 and hsa_circ_0001649 can be secreted from LC cell lines to the conditioned media, and that RT-ddPCR is able to detect the presence of these two circRNAs. Therefore, as the next step, we isolated RNA from the blood plasma and tested the presence of hsa_circ_0000190 by RT-ddPCR (Figure 3F,G). cDNA sample was serially diluted in ratio 1:2, and absolute quantification revealed linear amplification of hsa_circ_0000190 dependent on concentration (Figure 3G).

### 2.4. Expression of hsa_circ_0000190 and hsa_circ_0001649 in LC Patients with Different Stages and Tumor Sizes

As shown above, hsa_circ_0000190 demonstrated potential to predict the efficacy of lung cancer treatment. As human blood is the most commonly accessible specimen for diagnostic purposes, we performed serial blood analysis of the differential expression of hsa_circ_0000190 and hsa_circ_0001649 in a cohort of 231 LC patients and 41 healthy donors (Table 1). Intersection analysis of circRNA expression from serial blood tests was performed to evaluate the association between clinical characteristics and circRNA expression. First, we tested the presence of hsa_circ_0000190 and hsa_circ_0001649 in patients’ plasma samples, and showed that both circRNAs were detectable by RT-ddPCR with significantly higher levels of hsa_circ_0000190 than that of hsa_circ_0001649 (*p* < 0.0001) (Figure 4A). qRT-PCR was also performed for detecting plasma circRNAs; however, neither circRNA could be detected by this method (data not shown). The differences in the circRNA expression profiles between LC patients and healthy controls were also evaluated. Hsa_circ_0000190 was significantly upregulated in LC patients compared with healthy controls (*p* < 0.0001), and hsa_circ_0001649 was upregulated to a lesser extent (Figure 4B). Considering the low sensitivity or specificity of the current blood biomarkers of LC, there is a need for identification of new biomarkers in order to pursue more precise diagnosis and monitoring of the disease. The diagnostic accuracy of hsa_circ_0000190 was evaluated by the receiver operating characteristic (ROC) curve analysis. The best cut-off value for hsa_circ_0000190 expression threshold was calculated by the Youden’s index (sensitivity + specificity − 1) to maximize the sensitivity and specificity. Accordingly, we chose 2722 copies/mL as a cut-off value, at which Youden’s index was maximal. As shown in Figure 4C, a comparison between the stage I-IV lung cancer group and the healthy controls was performed, and the area under the curve of ROC (AUC) was 0.95 [95% confidence interval (CI) = 0.926–0.974; *p* < 0.0001] with PPV = 98.1% and NPV = 61.6%. Using 2722 copies/mL as a cut-off value for hsa_circ_0000190 expression threshold, the sensitivity and specificity of hsa_circ_0000190 for the diagnosis of LC were 0.9 and 0.902, respectively. In addition, the TNM stage subgroups were analyzed (Figure 4D,E), and the AUC of 0.96 in the group of stages III–IV (Figure 4E) was higher than the AUC of 0.896 in the group of stages I-II (Figure 4D). Altogether, these results indicate that the performance of hsa_circ_0000190 was better in the later stage (stage III-IV) of LC compared to that in the early stage (stage I-II). Patients with stage IV LC demonstrated significantly higher expression levels of hsa_circ_0000190 compared with patients of stages I-III (*p* < 0.0001) (Figure 4F), and the similar result was obtained when stages III-IV were compared with stages I-II patients (*p* < 0.0001) (Figure 4G). In addition, the primary tumor size was shown to moderately correlate with hsa_circ_0000190 (Figure 4H) and hsa_circ_0001649 (Figure 4I) expression levels (*p* < 0.0001).

### 2.5. Plasma Levels of Hsa_circ_0000190 Negatively Correlate with the Response to Immunotherapy

CircRNAs have been reported to be involved in a variety of molecular processes, including chemotherapeutic resistance. Evidence also exists for an association of circRNAs with immune checkpoint proteins, such as PD-1 and PD-L1 [30]. However, it is still unclear whether circRNAs can serve as monitoring markers correlating with cancer progression and reflecting drug treatment response or resistance in LC. Therefore, we further explored the potential of hsa_circ_0000190 and hsa_circ_0001649 to predict and monitor treatment responses in advanced LC patients undergoing immune therapy. Our study demonstrated that the level of hsa_circ_0000190 in plasma was related to distant metastasis (Figure 5A) and histology of adenocarcinoma (Figure 5B). Patients with 3–4 metastatic organs demonstrated higher levels of hsa_circ_0000190 than those with 1–2 metastatic organs (*p* = 0.0039) (Figure 5A). Lung adenocarcinomas with a micropapillary/solid-predominant pattern showed a higher level of hsa_circ_0000190 (*p* = 0.0028) than those with a non-micropapillary/solid-predominant pattern (including lepidic, acinar, or papillary patterns) (Figure 5B). The levels of hsa_circ_0000190 in plasma were also associated with extrathoracic metastasis (Figure 5C) and PD-L1 expression (Figure 5D). LC patients with extrathoracic metastasis demonstrated a higher level of hsa_circ_0000190 compared with those without (*p* = 0.0004). LC with positive PD-L1 expression (≥1%) had a higher level of hsa_circ_0000190 (*p* = 0.0283); however, in patients with negative PD-L1 expression, hsa_circ_0000190 could still be detected. The correlation coefficients between the expression level of PD-L1 and the level of hsa_circ_0000190 and hsa_circ_0001649 were 0.037 and 0.167, respectively. Patients with partial regression (PR) after treatment had lower levels of hsa_circ_0000190 compared with patients with stable disease (SD), progressive disease (PD), or SD/PD (*p* = 0.0036, 0.0001 and = 0.0002, respectively) (Figure 5E). Forty of 50 patients receiving immunotherapy were eligible for disease response evaluation, including 21 SD, 9 PD, 10 PR, and 0 complete response (CR). LC patients receiving immune-oncology (IO) therapy with CR/PR (responders) had lower levels of hsa_circ_0000190 compared with those with SD/PD (non-responders) (*p* = 0.0058) (Figure 5F). Seven of 10 patients receiving IO with PR tested negative for PD-L1 expression, so the hsa_circ_0000190 expression level is a potential biomarker for the effectiveness of LC immunotherapy in addition to PD-L1 expression. Using 2722 and 1681 copies/mL as cut-off values for the expression of hsa_circ_0000190 and hsa_circ_0001640, respectively, patients with a high level of hsa_circ_0000190 demonstrated significantly worse overall survival (*p* = 0.047) (Figure 5G), but those with a high level of hsa_circ_0001649 did not (Figure 5H). Collectively, our data suggest that the expression level of hsa_circ_0000190 could be a surrogate marker for response to LC treatment including immunotherapy.

### 2.6. Monitoring the Treatment Response by Plasma Levels of circRNAs in LC Patients Receiving Immunotherapy

The serial blood tests from lung cancer patients were also performed for evaluating the association between treatment response and the serial changes in the levels of hsa_circ_0000190 and hsa_circ_0001649 during the treatment period. The changes of plasma circRNA levels during immunotherapy treatment were monitored in six lung cancer patients (Figure 6). Three of these patients, who received atezolizumab or nivolumab, had low/negative expression of PD-L1 and progressive disease, demonstrated an obvious increase in the circRNA levels (Figure 6A–C). One 68 y/o female with advanced adenocarcinoma of the lung received atezolizumab after the failure of several lines of chemotherapy, and the serial blood tests showed an increase in the plasma levels of hsa_circ_0000190 and hsa_circ_0001649. She had progressive disease after treatment with atezolizumab (Figure 6A). A 59 y/o male with advanced adenocarcinoma of lung receiving nivolumab treatment demonstrated an increased level of both circRNAs upon disease progression. His chest CT scan revealed an increased tumor size after treatment (Figure 6B) and the similar finding was also noted in another 71 y/o male with advanced adenocarcinoma of the lung (Figure 6C). An 83 y/o male with advanced poorly differentiated NSCLC and positive PD-L1 expression (tumor proportion score [TPS] = 70%) had an obvious decrease in the plasma circRNA levels after receiving nivolumab, and significant improvement of symptoms was also noted (Figure 6D). His chest CT scan revealed a decreased tumor size following treatment and his reduced tumor burden after immunotherapy provided significant benefit to the patient and was accompanied by an obvious decrease in hsa_circ_0000190 (Figure 6D). A 67 y/o female with advanced adenocarcinoma of the lung received nivolumab after several lines of chemotherapy. This patient benefited from immunotherapy despite negative PD-L1 expression and concomitantly showed decreased expression of both circRNAs in serial blood tests (Figure 6E). Moreover, a similar finding was also revealed in another 79 y/o female with advanced adenocarcinoma of the lung without PD-L1 testing (Figure 6F). Overall, these data suggest that hsa_circ_0000190 and hsa_circ_0001649 may be utilized as novel biomarkers for the effectiveness of immunotherapy, regardless of the PD-L1 expression level.

### 2.7. Bioinformatics Analysis of Potential Downstream Network for Hsa_circ_0000190 and Hsa_circ_0001649 in LC

CircRNAs are known to work as competing endogenous RNAs (ceRNAs) that sequester miRNAs like a sponge, and hence regulate the stability or translation of target mRNAs by alleviating them from the miRNA-mediated inhibition [31]. To analyze the potential downstream regulatory network of hsa_circ_0000190 and hsa_circ_000164, we identified the predicted miRNA targets of these circRNAs using the Circular RNA Interactome database (http://circinteractome.nia.nih.gov): the former contained the binding sites of 12 miRNAs and the latter of 31 miRNAs (Figure 7A, Appendix A). We performed qRT-PCR to analyze the expression predicted miRNAs in BEAS-2B and HCC827 cell lines, and found that four of hsa_circ_0000190 target miRNAS and three of hsa_circ_000164 target miRNAs were drastically downregulated in HCC827 cell line as compared to BEAS-2B (Figure 7B). The targets of hsa_circ_0000190 were miR-767-5p, miR-382-5p, miR-382-3p, and miR-1299 and the targets of hsa_circ_000164 were miR-942-5p, miR-889-3p, and miR-140-3p. All of them contained one binding site within their respective circRNAs (Figure 7C). Given that a typical sponge circRNA, like CDR1as that targets miR-7 [32], contains multiple miRNA binding sites, we speculate that hsa_circ_0000190 and hsa_circ_000164 may regulate their target miRNAs by recruiting the RNA degradation machinery, thus actively decreasing their levels. The top three downregulated miRNAs from each group were then subjected to further downstream network analysis. From our RNA-Seq data, we identified that some of the top mRNA targets of the downregulated miRNAs targeted by hsa_circ_0000190 and hsa_circ_000164, were highly upregulated in HCC827 cell line compared to BEAS-2B (Figure 7A, Appendix A) and were organized in a functional network (Appendix A). Interestingly, miR-1299 has been reported to be a tumor suppressor that inhibits cell proliferation and STAT3 pathway [33,34,35]; whereas miR-767 has been reported to affect the maintenance of normal lung function [36].

## 3. Discussion

Advanced LC is often characterized by drug resistance-related treatment failure. The role of miRNAs, a type of small non-coding RNAs (ncRNAs), as prognosis predictors for LC and other cancer types has been explored by previous studies, but the developed miRNA panels use relative miRNA ratio measurement [37,38,39]. More recently, growing scientific studies continue to associate circRNAs with oncogenesis, metastasis and drug resistance in LC [27,28,40,41]. For instance, circRNA_100876 was described as a prospective LC biomarker by Liao et al. [27]. Likewise, hsa_circ_0013958 and circRNA_102231 were also reported to be upregulated in lung adenocarcinoma patients [11,28]. Of note, all the highlighted studies used relative quantification of qRT-PCR data in their analysis. However, it still an open question whether relative quantification method can be appropriate for real-time detection of circRNAs from the limited amount of LB samples in order to monitor tumor status, especially in patients with treatment failure-associated advanced LC.

In this study, in order to detect circRNAs secreted into blood plasma, we applied highly sensitive and specific RT-ddPCR method with absolute quantification. By using it, we discovered that the levels of hsa_circ_0000190 and hsa_circ_0001649 were increased in LC cell lines and patients’ plasma. These two circRNAs were shown to be secreted by LC cells, as their presence was detected in the conditioned media of LC cell lines. Furthermore, we propose that RT-ddPCR can be used to analyze the plasma levels of hsa_circ_0000190 and hsa_circ_0001649 in patients’ blood. Our result showed that the plasma level of hsa_circ_0000190 was related to clinicopathological features, treatment response, and overall survival in LC patients. Accordingly, hsa_circ_0000190 was demonstrated to be a potentially valuable blood-based biomarker to assess the prognosis of LC and the treatment efficacy of immunotherapy by ddPCR-based LB.

Due to their circular nature, circRNAs are highly stable RNA species, which makes them excellent candidates for biomarkers detected in the nuclease-rich blood environment [13,17,27]. For example, Zhu et al. found an association between upregulation of circRNA 100876 with lymph node metastasis in NSCLC [27]. Hsa_circ_0013958 was characterized as a miR-134 sponge, leading to upregulated oncogenic cyclin D1 in lung adenocarcinoma [28]. Cinnamaldehyde-induced cell apoptosis was also demonstrated to be mediated by the hsa_circ_0043256/miR-1252/ITCH axis, providing new insight into LC molecular mechanisms [40]. Similarly, our data demonstrate that the expression of hsa_circ_0000190 in the plasma specimens was consistent with its upregulated expression in LC cell lines. Our in vitro experiments indicated that LC cells may secrete hsa_circ_0000190 and hsa_circ_0001649 into the plasma (Figure 3). Indeed, the levels of hsa_circ_0000190 and hsa_circ_0001649 in the plasma of pre-treated LC patients were higher than those in control healthy individuals (Figure 4B). The sensitivity of hsa_circ_0000190 for the diagnosis of LC is lower in the group of early stages (I–II) than late stages (III–IV) (0.815 vs. 0.904) (Figure 4D,E). Although hsa_circ_0000190 may be less sensitive for early stage NSCLC, its specificity for the diagnosis of LC is higher in the group of early stages than late stages (0.976 vs. 0.927). Therefore, it shows the potential ability to serve as a biomarker for improving selection of patients for lung cancer screening, and to estimate the likelihood of malignancy in pulmonary nodules identified at screening. Combining low-dose CT and potential biomarkers with optimal discriminating power in high risk populations for lung cancer may optimize the effectiveness for the early detection of lung cancer [42].Lung adenocarcinoma with micropapillary and/or solid patterns has been reported to have the worst prognosis, and in our study, the samples from patients with this type of LC showed higher levels of hsa_circ_0000190 (*p* = 0.0028) (Figure 5B). This result indicates that a higher level of hsa_circ_0000190 is positively correlated with histological types associated with a worse prognosis [43]. The average copy numbers of hsa_circ_0000190 gradually decreased after patients received effective therapy (Figure 6D–F). In our serial blood examinations, the copy number change in hsa_circ_0000190 in patients receiving treatment was identified as an independent monitoring indicator for the progression of LC (Figure 6). The hsa_circ_0000190 level in the plasma of LC patients was significantly correlated with treatment response in the eligible LC patients (Figure 5E,F). These results suggest that hsa_circ_0000190 may be utilized as a novel LC-associated biomarker for the evaluation of tumor progression. In addition to the prospective LC biomarker or molecular identification, our results also suggest that circRNA has the potential ability to predict treatment responses to immunotherapy.

Immunotherapy through immune checkpoint inhibition by blocking the PD-1/PD-L1 signaling pathway that enhances anti-cancer T cell immunity has shown promising and significant efficacy in a variety of malignancies, including LC [44,45,46,47]. Clinical trials of PD-1/PD-L1 blocking antibodies demonstrated that although PD-L1 expression levels were associated with a positive response to immunotherapy, many tumors without PD-L1 expression still responded to the treatment [45,47]. Accordingly, PD-L1 expression may not be the optimal and solitary biomarker to predict the immunotherapy efficacy [45]. The identification of improved biomarkers that predict which patients will respond to immunotherapy is an urgent need.

In this study, hsa_circ_0000190 showed higher expression in LC with positive PD-L1 expression (*p* = 0.0283) (Figure 5D). A previous report revealed that hsa_circ_0020397 could inhibit the function of miR-138 leading to upregulation of miR-138 targets, including PD-L1 in colorectal cancer [30]. In our case, the effect of hsa_circ_0000190 on PD-L1 expression is still unclear. The detailed mechanism of interplay between circRNA and anti-tumor immunity remains to be elucidated. Moreover, the potential biological functions identified by bioinformatics analysis of the two circRNAs still need further characterization to delineate the role and regulatory mechanism of hsa_circ_0000190 and hsa_circ_0001649, as these circRNAs may not be only biomarkers in patients’ liquid biopsy, but could also be involved in tumorigenic molecular pathways. Further investigation on future perspectives of clinical application are mandatory. LC patients with response to immunotherapy had lower levels of hsa_circ_0000190 compared with those without response (*p* = 0.0058) (Figure 5F). Seven of 10 patients receiving IO with PR lacked PD-L1 expression; therefore, the hsa_circ_0000190 expression level is a potential biomarker for LC immunotherapy in addition to PD-L1 expression. To facilitate the development of immunotherapy, further studies are required to validate the predictive value of circRNAs for the treatment response to LC immunotherapy.

Summarily, in this study, we used LB-based ddPCR platform to demonstrate that the plasma level of hsa_circ_0000190 can be monitored and may serve as a useful blood-based biomarker to monitor the disease status and the treatment efficacy. In terms of cost, dye-based ddPCR is more economical than its probe-based counterpart, making it more adaptable to an average molecular laboratory setting.

## 4. Methods

### 4.1. Patient Population

Lung cancer (LC) patients who had received diagnosis, staging, and treatment were recruited into the present study. Patients were excluded if they had been diagnosed with other cancers; had incomplete medical records or had received less than 3 months of follow-up. This study enrolled 272 cases, including 231 LC patients and 41 healthy controls. Of 231 patients, 156 (67.5%) were men and 75 (32.5%) were women (Table 1). The histologic types included 195 adenocarcinoma (84.4%), 19 squamous cell carcinoma (8.2%), 2 small cell carcinoma (0.9%), and 15 others (6.5%). Among them, 65 (28.1%) had stage I-II disease and 166 (71.9%) had stage III–IV disease; 144 (62.3%) had distant metastases, while 87 (37.7%) patients did not. Fifty (21.6%) patients received immunotherapy treatment, including programmed death 1 (PD-l) and programmed death-ligand 1 (PD-L1) blocking antibodies. A total of 133 patients receiving systemic treatment were eligible for evaluation of treatment response. Of those, 70 (30.3%) had an *EGFR* mutation (27 exon 19 deletions, 38 exon 21 L858R) (Table 1). The characteristics of 166 stage IIIA–IV lung cancer patients who received systemic treatment, including immunotherapy, chemotherapy, and targeted therapy, are shown in Appendix A. This study was approved by the Institutional Review Board of Taipei Veterans General Hospital. The LC stage was evaluated according to the seventh edition of the tumor node metastasis staging system, and the histological classification was assessed following the 2015 World Health Organization (WHO) classification of lung tumors.

Serial blood specimens were collected from patients before and after treatment. All specimens were collected in Vacutainer EDTA tubes (Becton Dickinson, Franklin Lakes, NJ, USA) and centrifuged for 10 min at 1000× *g*, no later than 30 min after the collection of specimens for harvesting plasma under room temperature. Plasma was stored at −80 °C in 1 mL aliquots until use and analyzed within 2 days. All procedures of tissues acquirements have followed the tenets of the Declaration of Helsinki and are reviewed by Institutional Review Committee at Taipei Veterans General Hospital.

### 4.2. Cell Culture

Human lung cancer cell lines [48] were obtained from the American Type Culture Collection (ATCC) before 2007 and tested positive for human origin. Cells were cultured in DMEM with 10% fetal bovine serum, 100 units/mL penicillin, and 100 μg/mL streptomycin under standard culture conditions (37 °C, 95% humidified air, and 5% CO_2_). Subculturing was performed using trypsin-EDTA. The medium was refreshed every two days. All cells lines were tested negative for mycoplasma contamination.

### 4.3. RNA Isolation and qRT-PCR

Total RNA was isolated from human lung cancer cells using the RNeasy Mini Kit (QIAGEN, Hilden, Germany). Oligonucleotides were designed using the computer software package Primer Express 2.0 (Applied Biosystems, Foster City, CA, USA). All oligonucleotides were synthesized by Invitrogen (Carlsbad, CA, USA). Oligonucleotide specificity was computationally tested by homology search with the human genome using BLAST (National Center for Biotechnology Information, Bethesda, MD, USA) and later confirmed by dissociation curve analysis. The real-time quantitative polymerase chain reaction (qRT-PCR) was performed using the SYBR Green method in an ABI 7000 sequence detection system (Applied Biosystems) per the manufacturer’s guidelines. The relative quantification of circRNA expression levels in qRT-PCR was evaluated by the ΔCq method. The sequences of divergent primers to amplify the spice junction of hsa_circ_0000190 were 5′-TTGCTCCTTGGGCGCTATAC-3′ and 5′-AGAGTCCAGCGGCAAAACTA-3′ [49]. The sequences of divergent primers for hsa_circ_0001649 were 5′-AATGCTGAAAACTGCTGAGAGAA-3′ and 5′-TTGAGAAAACGAGTGCTTTGG-3′ [50]. The convergent primers to amplify glyceraldehyde 3-phosphate dehydrogenase (*GAPDH*) were 5′-GCATTGCCCTCAACGAC-3′ and 5′-GTCTCTCTCTTCCTCTTGTGC--3′. These primers were synthesized by Invitrogen. The gradient dilution was performed to determine their suitable concentration.

### 4.4. qRT-PCT and RT-ddPCR for Detecting Plasma circRNA

Blood specimens were collected in Vacutainer EDTA tubes (Becton Dickinson) and processed as per the above-described protocol. Total RNA was isolated from 1 mL of plasma by using the QIAamp Circulating Nucleic Acid Kit (QIAGEN). The RNA was eluted from spin columns in 15 μL of nuclease-free water. Following the manufacturer’s instruction for plasma specimens, the cDNA Synthesis Kit (Thermo Fisher Scientific, Waltham, MA, USA) was used to synthesize complementary DNA (cDNA) in a 20 μL reaction, starting from 3 μL of RNA. Then, 1 μL of synthesized cDNA was assayed in a 20 μL PCR reaction volume according to the manufacturer’s protocol. The ddPCR was processed on the QX200 Droplet System with ddPCR EvaGreen Supermix (Bio-Rad, Hercules, CA, USA) and 1 μL of synthesized cDNA. The droplets for the PCR reactions were generated according to the manufacturer’s protocol (Bio-Rad). Then, the samples were transferred into a 96-well PCR plate and the ddPCR was conducted. The above-mentioned procedure was performed for all test specimens and negative controls. At the end of the PCR reaction, PCR-positive and PCR-negative droplets were counted by the QX200 Droplet Reader (Bio-Rad) and the data were analyzed by QuantaSoft software (Bio-Rad).

### 4.5. RNA-Seq

For NGS data analysis, we first used PEAT algorithm to remove adapter contamination and then aligned the clean reads to the mm10 genome using RNA-Star (version 2.5.3a) to retain junction reads before sending into Cufflinks (version 2.2.1) with gene annotation by GENCODE (version 21) for normalized expression level estimation (i.e., FPKM). We only considered circRNAs, miRNAs, and protein-coding mRNAs in this work. The Log_2_ fold change of the FPKM value of each gene between samples was used to identify up- or down-regulated genes over the control. The PCA analysis was applied to several samples with all the FPKM values as features and three top principal components were extracted and used as the new basis to further discover the similarity between samples. The RNA-Seq data were deposited to the Gene Expression Omnibus (GEO) database with the accession number GSE152434.

### 4.6. Evaluation of LC Treatment Efficacy

The baseline evaluations of the LC characteristics for each patient were performed within the 3 weeks prior to treatment. This included a chest computed tomography (CT) scan, which was repeated every 3 months thereafter or when confirmation of treatment response or disease progression was needed. Treatment response assessment was performed according to the Response Evaluation Criteria in Solid Tumors (RECIST) group criteria (version 1.1). Progression-free survival (PFS) was calculated from the date of treatment initiation to the earliest sign of disease progression, as determined by the RECIST criteria, or death from any cause. If disease progression had not occurred at the time of the last follow-up visit, the data on PFS was censored at that time. Overall survival was measured from the date of treatment initiation until the date of death or last follow-up.

### 4.7. Statistical Analysis

Quantifiable data are expressed as mean ± standard error of the mean (SEM). Differences between the groups were analyzed using one-way ANOVA followed by Student’s *t*-test. The Kaplan–Meier method with a log-rank test was used for survival analysis. When comparing the treatment response and biomarkers, the Mann–Whitney test was used for non-parametric data and Pearson’s χ^2^ test was used for parametric data. Statistical analysis was performed using the Statistical Package for the Social Sciences (SPSS) 18.0 Software (SPSS, Chicago, IL, USA) and PRISM (GraphPad Software Inc., San Diego, CA, USA). All *p*-values were 2-sided, and a value of <0.05 was considered statistically significant.

## 5. Conclusions

In conclusion, our research is the first to determine the absolute cell-free circRNA profile in the plasma of patients with LC. Furthermore, we discovered that hsa_circ_0000190 and hsa_circ_0001649 levels increased in lung cancer cell lines and patients’ plasma. In addition, the level of hsa_circ_0000190 was associated with clinicopathological features and the treatment response of LC patients. As a result, hsa_circ_0000190 may be a valuable blood-based biomarker to estimate the prognosis of LC and the treatment efficacy of immunotherapy by ddPCR-based LB. We demonstrated that LB-based circRNA-ddPCR systems serve as a platform of personal precision medicine—prospectively validating the efficacy of chemotherapy, targeted remedy, and immune therapeutics in advanced LC.

## Figures and Tables

**Figure 1 cancers-12-01740-f001:**
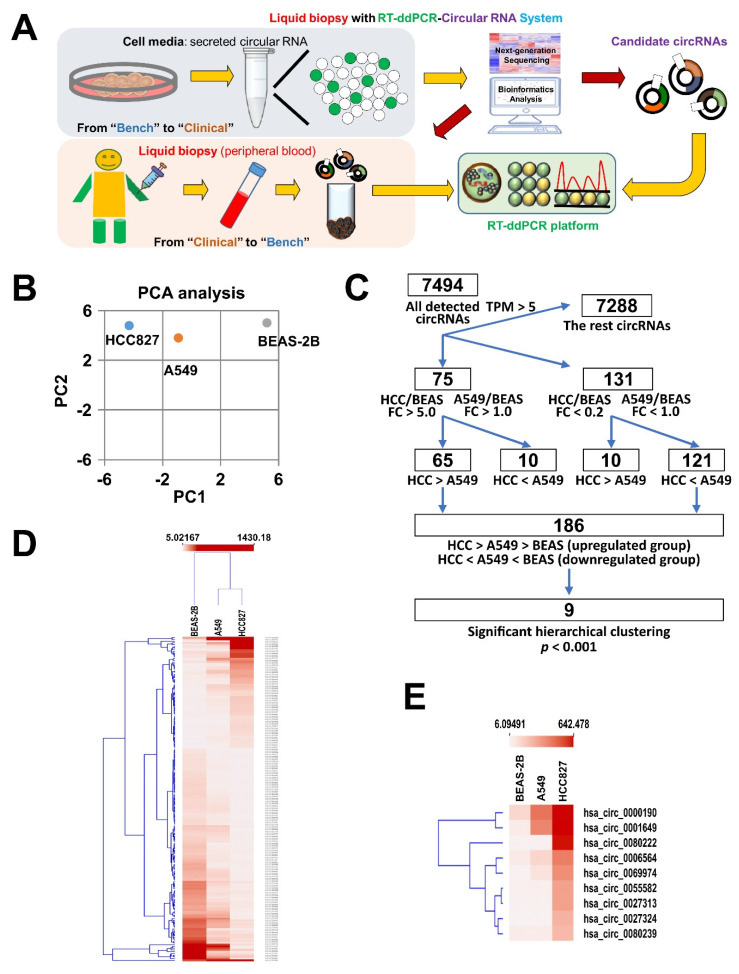
Identification of potential secretory circRNA biomarker candidates using RNA-Seq. (**A**) Descriptive flow diagram of experimental plan to identify potential circRNA biomarkers of LC. (**B**) Principal component analysis (PCA) of circRNA sequencing results of LC cell lines (HCC827 and A549) and normal bronchial epithelial cells (BEAS-2B). (**C**) Flow diagram of RNA-Seq analysis aimed at selecting the best candidates of circRNA LC biomarkers. (**D**,**E**) Hierarchical clustering heatmap of 189 circRNA candidates (**D**) identified according to the pipeline shown in (**C**) and a subset of 9 circRNAs with the most prominent fold changes (**E**).

**Figure 2 cancers-12-01740-f002:**
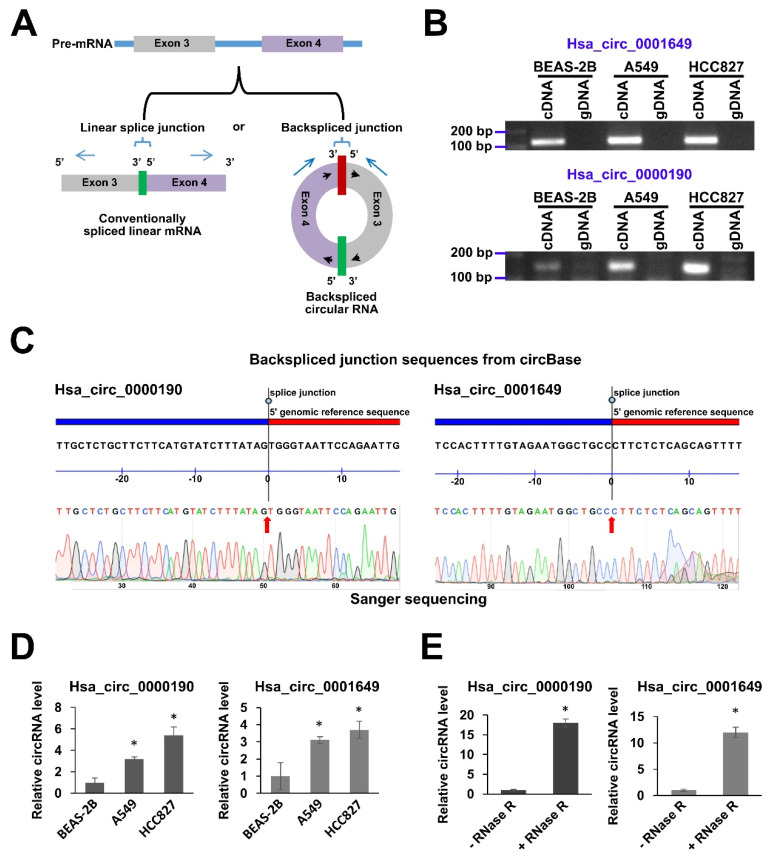
Validation of expression of hsa_circ_0000190 and hsa_circ_0001649 in LC cell lines. (**A**) Schematic illustration of a circRNA and its linear form counterpart. The divergent primers (blue arrows) across the backspliced junction can only amplify the circRNA form. (**B**) Agarose gel electrophoresis analysis of the sizes of RT-PCR amplicons of hsa_circ_0000190 and hsa_circ_0001649 amplified from cDNA templates of the indicated cell lines. gDNA templates were used as a negative control. (**C**) Sanger sequencing of RT-PCR amplicons of hsa_circ_0000190 and hsa_circ_0001649 (**B**) aligned to the respective sequences of these circRNAs retrieved from the circBase database. (**D**) qRT-PCR analysis of expression of hsa_circ_0000190 and hsa_circ_0001649 in BEAS-2B, A549 and HCC827 cell lines. The results expressed as fold change relative to BEAS-2B. (**E**) qRT-PCR analysis of expression of hsa_circ_0000190 and hsa_circ_0001649 in A549 cells, with or without prior treatment of total RNA with RNase R. The results expressed as fold change relative to—RNase R. In (**D**) and (**E**), fold changes were calculated using ddCt method with *GAPDH* mRNA used as a normalization control. Mean fold changes from three independent experiments are shown with standard deviation error bars. * *p* < 0.001 (Student’s *t*-test).

**Figure 3 cancers-12-01740-f003:**
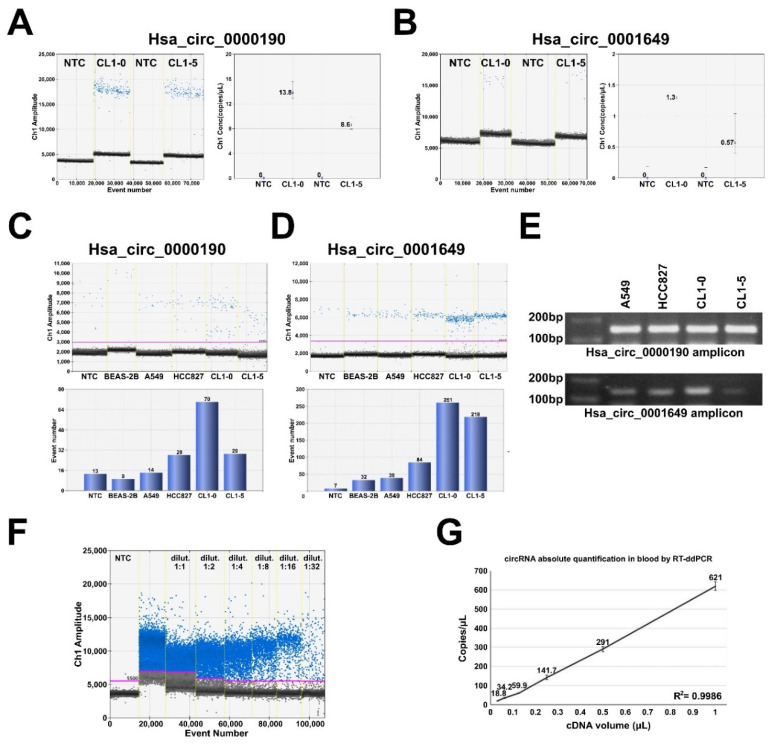
RT-ddPCR detection of hsa_circ_0000190 and hsa_circ_0001649 secreted by LC cell lines and in human blood plasma. (**A**,**B**) ddRT-PCR analysis of the expression of hsa_circ_0000190 (**A**) and hsa_circ_0001649 (**B**) in the medium conditioned by CL1-0 and CL1-5 cell lines. Left—amplification plot, right—absolute quantification. NTC—no template control. (**C**,**D**) ddRT-PCR analysis of the expression of hsa_circ_0000190 (**C**) and hsa_circ_0001649 (**D**) in the medium conditioned by BEAS-2B and the indicated LC cell lines. Top—amplification plot, bottom—quantification. (**E**) Agarose gel electrophoresis analysis of the sizes of RT-PCR amplicons of hsa_circ_0000190 and hsa_circ_0001649 amplified from cDNA templates of the indicated cell lines. (**F**,**G**) ddRT-PCR analysis of the expression of hsa_circ_0000190 in the plasma sample. cDNA was diluted at the indicated ratios, (**F**)—amplification plot, (**G**)—absolute quantification.

**Figure 4 cancers-12-01740-f004:**
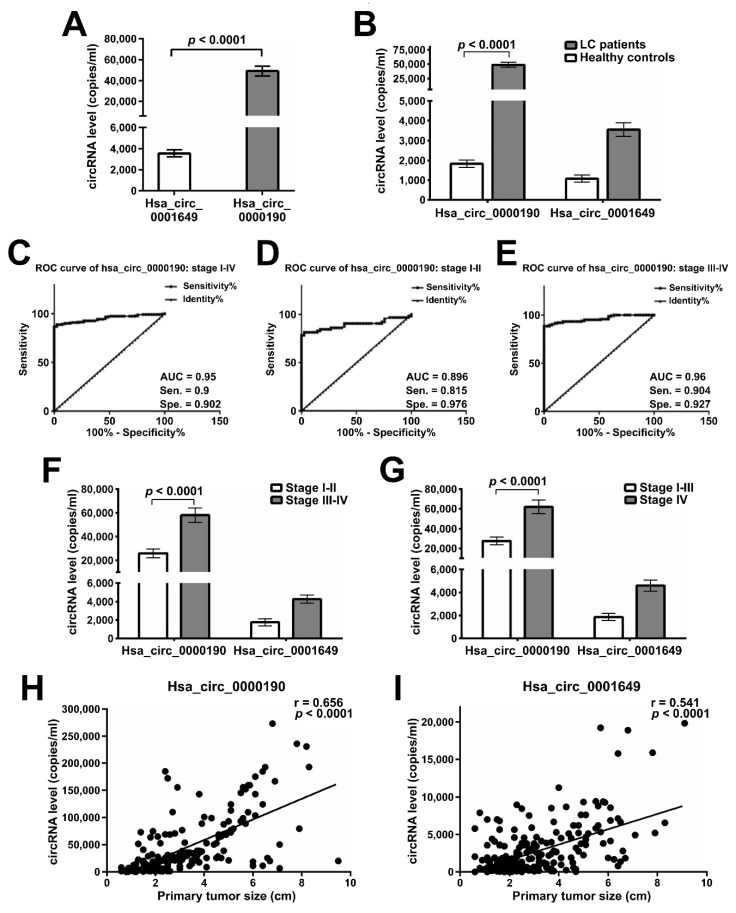
Expression levels of plasma circRNAs in different stages and tumor sizes of LC. (**A**) RT-ddPCR analysis of expression of hsa_circ_0000190 and hsa_circ_0001649 in the plasma of lung cancer patients’ blood samples. Mean values are shown with standard deviation error bars, *p*-value calculated by Student’s *t*-test. (**B**) RT-ddPCR analysis of expression of hsa_circ_0000190 and hsa_circ_0001649 in the plasma of lung cancer patients’ and healthy controls’ blood samples. Mean values are shown with standard deviation error bars, *p*-value calculated by Student’s *t*-test. (**C**–**E**) ROC curve analysis of diagnostic ability of hsa_circ_0000190 to discriminate between healthy individuals and LC patients of all TNM stages (**C**), stages I-II (**D**), stages III-IV (**E**). 2722 copies/mL was used as a cut-off value for hsa_circ_0000190 expression threshold. AUC—area under the curve, Sen.—sensitivity, Spe.—specificity. (**F**,**G**) RT-ddPCR analysis comparing the expression of hsa_circ_0000190 and hsa_circ_0001649 in the plasma of lung cancer patients of stages I-II and stages III-IV (**F**), stages I-III and stage IV (**G**). Mean values are shown with standard deviation error bars, *p*-value calculated by Student’s *t*-test. (**H**,**I**) Pearson correlation analysis of expression levels of hsa_circ_0000190 (**H**) and hsa_circ_0001649 (**I**) and primary tumor sizes.

**Figure 5 cancers-12-01740-f005:**
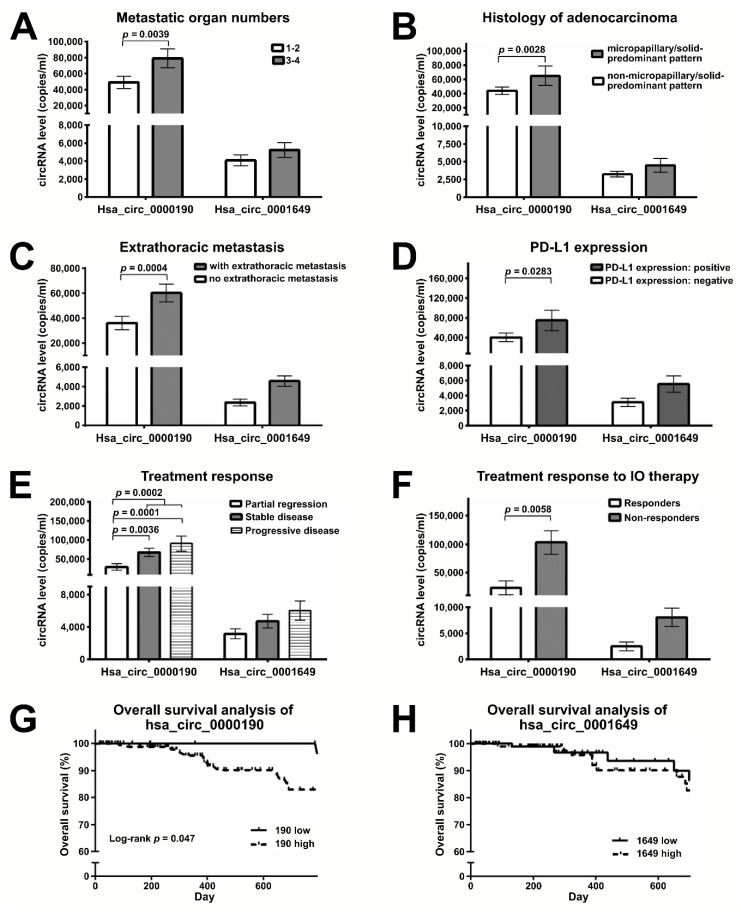
Plasma levels of hsa_circ_0000190 negatively correlate with the response to immunotherapy. (**A**–**F**) RT-ddPCR analysis comparing the expression of hsa_circ_0000190 and hsa_circ_0001649 in the plasma of lung cancer patient groups with: (**A**) different numbers of metastatic organs (1–2 vs. 3–4 organs); (**B**) micropapillary/solid-predominant histological type vs. other histological types; (**C**) with vs. without extrathoracic metastasis; (**D**) positive vs. negative PD-L1 expression; (**E**) the indicated types of treatment response; (**F**) positive vs. negative response to IO therapy. Mean values are shown with standard deviation error bars, *p*-value calculated by Student’s *t*-test. (**G**,**H**) Survival analysis of patients with high and low expression of hsa_circ_0000190 (G) and hsa_circ_0001649 (**H**).

**Figure 6 cancers-12-01740-f006:**
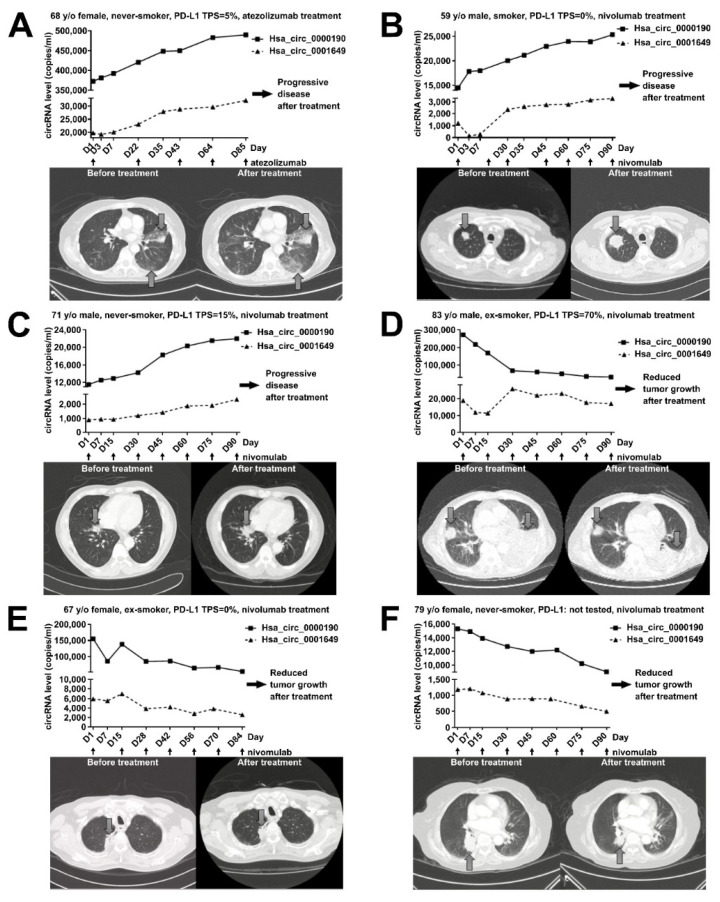
Monitoring the treatment response by plasma levels of circRNAs in lung cancer patients receiving immunotherapy. (**A**–**F**) RT-ddPCR analysis of expression of plasma hsa_circ_0000190 and hsa_circ_0001649 (top panel) and chest CT scans before and after treatment (bottom panel) in six cases of LC patients receiving immunotherapy. Black arrows indicate the time points of drug administration (atezolizumab or nivomulab) in the treatment time course. Big gray arrows indicate tumors on the CT scans.

**Figure 7 cancers-12-01740-f007:**
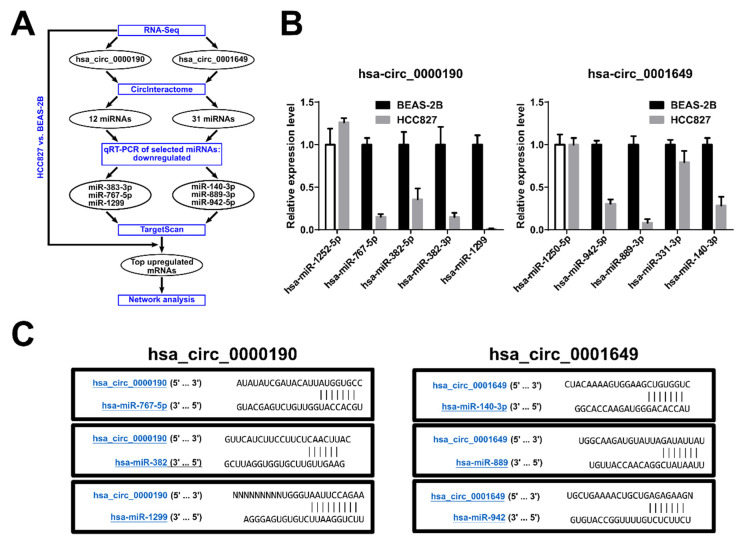
Bioinformatics analysis of potential downstream network for hsa_circ_0000190 and hsa_circ_0001649 in LC. (**A**) Pipeline of analysis. (**B**) qRT-PCR analysis of expression of the selected predicted miRNA targets of hsa_circ_0000190 (left) and hsa_circ_0001649 (right). (**C**) The predicted interaction sites between hsa_circ_0000190, hsa_circ_0001649 and predicted target miRNAs.

**Table 1 cancers-12-01740-t001:** Characteristics of Cases.

□	Total No.	*p*-Value	
		Hsa_circ_ 0000190	Hsa_circ_ 0001649
Patient no.	231	<0.0001 *	0.938
Healthy controls	41		
Gender (%)			
Male	156 (67.5)	0.213	>0.999
Female	75 (32.5)		
Mean age (range, year)	63.8 (35–90)	0.552	>0.999
With smoking history	110	0.922	0.997
Performance status (ECOG) (%)			
0	118 (51.1)	0.489	0.335
1	95 (41.1)		
2	18 (7.8)		
Lung cancer stage (%)			
I	62 (26.8)	I-II vs. III-IV <0.0001 *	I-II vs. III-IV 0.924
II	3 (1.3)		
IIIa	11 (4.8)		
IIIb	11 (4.8)		
IV	144 (62.3)		
Histology (%)			
Adenocarcinoma	195 (84.4)	0.386	0.915
Squamous cell carcinoma	19 (8.2)		
Non-adeno/non-sqcc NSCLC	15 (6.5)		
SCLC	2 (0.9)		
Primary tumor size (range, cm)	3.1 (0.6–9.5)		
Metastatic organs, mean (range)	1 (0–4)	<0.0001 *	0.899
Extrathoracic metastasis	125 (44.7)	0.0004 *	0.93
Positive PD-L1 expression (negative)	33 (56)	0.0283 *	0.981
Immunotherapy	50	Responder vs. non-responder	
Pembrolizumab	14	0.0058 *	0.972
Nivolumab	24		
Atezolizumab	9		
Durvalumab	3		
Systemic treatment response			
Complete remission (CR)	0	CR/PR vs. SD/PD 0.0002 * > 0.999	
Partial remission (PR)	37		
Stable disease (SD)	69		
Progressive disease (PD)	27		
*EGFR* mutation		*EGFR* mutation vs. wild type	
with any mutation	73	0.243	0.987
with exon 19 deletion	27		
with L858R mutation	38		
Wild type	95		

* Significant association. ECOG: Eastern Cooperative Oncology Group; adeno: adenocarcinoma; sqcc: squamous cell carcinoma.

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
