# Peer review of "Plasma Level of Circular RNA hsa_circ_0000190 Correlates with Tumor Progression and Poor Treatment Response in Advanced Lung Cancers"

_cancers, 2020, doi:10.3390/cancers12071740_

Round 1
Reviewer 1 Report
Authors of this study have systematically evaluated the role of circular RNAs for the prognostic application of lung cancer. They have successfully demonstrated a direct correlation of two circular RNA expression with tumor progression and an inverse correlation of their expression levels to that of poor response to anti-cancer therapy. Authors have also bioinformatically predicted the possible downstream miRNA/mRNA targets of those circular RNAs to provide mechanistic direction. However, there are several limitations that are listed below which need to be addressed to make this study complete.
- Authors need to mention how many miRNA binding sites are there in each circular RNA. In fact, this number of binding sites should be used as one of the factors for filtering out the circular RNA candidates for functional validation along with the fold change and significance parameters.
- The major drawback in the study is that it lacks to provide target validation to the identified circular RNA molecules. Authors need to perform loss or gain of function studies using siRNAs against circular RNAs. They should also perform independent experiment wherein they silence the corresponding miRNAs using antagomiRs to see which mRNAs from their prediction is affected.
- (CHECK) Also, authors can check the status of those predicted target mRNAs of the circular RNAs from their RNA sequencing data.
- Authors need to perform similar bioinformatics analysis to show the downregulated circular RNAs to their upregulated miRNA and downregulated mRNA targets.
- Authors need to explain how early detection of the NSCLC is possible when the circular RNAs are easily detectable only at the later stages of the cancer progression as shown in the results?
- Authors need to make the sequencing data available to the readers by depositing it in public repositories.
Author Response
Authors of this study have systematically evaluated the role of circular RNAs for the prognostic application of lung cancer. They have successfully demonstrated a direct correlation of two circular RNA expression with tumor progression and an inverse correlation of their expression levels to that of poor response to anti-cancer therapy. Authors have also bioinformatically predicted the possible downstream miRNA/mRNA targets of those circular RNAs to provide mechanistic direction. However, there are several limitations that are listed below which need to be addressed to make this study complete.
1. Authors need to mention how many miRNA binding sites are there in each circular RNA. In fact, this number of binding sites should be used as one of the factors for filtering out the circular RNA candidates for functional validation along with the fold change and significance parameters.
Reply:
Dear Reviewer. Thank you for your suggestion on our bioinformatics analysis. Accordingly, we completely modified the “Bioinformatics Analysis of Potential Downstream Network for Hsa_circ_0000190 and Hsa_circ_0001649 in LC” paragraph and modified the Figure 7. According to your suggestion, we specify the number of the identified miRNA targets of these circRNAs. We used qRT-PCR to analyze the levels of these miRNAs and found 3 most drastically downregulated in LC cell line as compared to normal BEAS-2B cells for further signaling network prediction. We speculate that circRNAs can trigger degradation mechanisms, not merely working as a sponge, but this is beyond the scope of this paper.
We want to emphasize that the major point of this paper is not identification of the detailed molecular mechanisms of how circRNAs 190 and 1649 work, but their potential usability as blood biomarkers for lung cancer. We present these predicted downstream network data, and we presume it can be used as a trigger for future studies. We don’t claim they are statistically valid, but we present an observation as a side aspect of this paper. We appreciate your comments that motivate us to polish our bioinformatics analysis presented in the manuscript.
2. The major drawback in the study is that it lacks to provide target validation to the identified circular RNA molecules. Authors need to perform loss or gain of function studies using siRNAs against circular RNAs. They should also perform independent experiment wherein they silence the corresponding miRNAs using antagomiRs to see which mRNAs from their prediction is affected.
Reply:
Thank you for this valid suggestion. Indeed, we agree that the proposed additional experiments would be of interest to pursue. However, the major point of this research was more related to the potential value to the identified circRNAs in reflecting patients disease status and treatment response, and that these circRNA could be detected by liquid biopsy/ddPCR, rather than dissecting detailed molecular mechanisms of their action. Due to time limitation given for revision, and fulfilment of the major objective of this study, we will consider the proposed experiments in our future research. Meanwhile, we have added Supplemental Table 2 and 3 to show the expression levels of the predicted mRNA targets from our RNA-seq, and the revised network maps in Supplemental Fig. 2 were colored to reflect the RNA-seq expression levels of each predicted downstream mRNA targets. We are currently establishing the overexpression and knockdown system for circRNA and hope that we can present the downstream molecular mechanisms of the two circRNA in near future. We still believe that in silico prediction of regulatory networks controlled by biomarker circRNAs briefly presented in this paper can be of interest to the scientific community and can serve as a trigger for future investigations.
3. Authors need to perform similar bioinformatics analysis to show the downregulated circular RNAs to their upregulated miRNA and downregulated mRNA targets.
Reply:
Thank you for this valuable suggestion. We definitely should use our RNA-Seq data to perform this kind of analysis in the future to identify potential circRNAs with tumor suppressor properties. However, we believe it is beyond the scope of this study, as its major focus was on identification of prognostic biomarkers for lung cancer, therefore, we narrowed down our analysis to upregulated circRNAs only.
4. Authors need to explain how early detection of the NSCLC is possible when the circular RNAs are easily detectable only at the later stages of the cancer progression as shown in the results?
Reply:
Thank you for this valuable comment. We have added the following description in the Discussion section to address this issue: “The sensitivity of hsa_circ_0000190 for the diagnosis of LC is lower in the group of early stages (I-II) than late stages (III–IV) (0.815 vs. 0.904) (Figure 4D-E). Although hsa_circ_0000190 may be less sensitive for early stage NSCLC, its specificity for the diagnosis of LC is higher in the group of early stages than late stages (0.976 vs. 0.927). Therefore, it shows the potential ability to serve as a biomarker for improving selection of patients for lung cancer screening, and to estimate the likelihood of malignancy in pulmonary nodules identified at screening. Combining low-dose computed tomography and potential biomarkers with optimal discriminating power in high risk populations for lung cancer may optimize the effectiveness for early detection of lung cancer.”
5. Authors need to make the sequencing data available to the readers by depositing it in public repositories.
Reply:
We have deposited RNA-Seq data to the GEO database, which is accessible with the accession number GSE152434. This information has been added to the Materials and Methods section.

Reviewer 2 Report
In this manuscript, authors have aimed to identify potential circRNA biomarkers of lung cancer using the liquid biopsy-based RT-ddPCR. Authors have used Next generation sequencing (NGS) to identify circRNAs overexpressed in lung cancer cell lines with different EGFR mutation status. Authors have shown two circRNAs, hsa_circ_0000190 and hsa_circ_0001649, secreted by LC cells in vitro, which were also detected in blood by RT-ddPCR. Further, they were able to monitor the expression of these two circRNAs in blood plasma of a cohort of LC patients, and correlated these circRNAs with a number of pathological parameters and response to immunotherapy. Authors conclude that circRNAs could be used as secreted biomarkers.
Overall, the manuscript is well-written. Methodology seems appropriate. Manuscript presents some interesting findings based on both in vitro and patient data. However, there are some minor corrections required in the manuscript which are provided below.
- Line 149, authors have cited Figure 1F. However, they have not provided this figure. Authors are suggested to provide expression levels of hsa_circ_0000190 and hsa_circ_0001649 in lung cancer cell lines.
- Line 164, Figure 1A should be given as Figure 2A.
- Authors should provide details of inclusion and exclusion criteria for patient population in the method section.
- Authors should provide details regarding cell line authentication and mycoplasma testing in cell culture method section.
Author Response
In this manuscript, authors have aimed to identify potential circRNA biomarkers of lung cancer using the liquid biopsy-based RT-ddPCR. Authors have used Next generation sequencing (NGS) to identify circRNAs overexpressed in lung cancer cell lines with different EGFR mutation status. Authors have shown two circRNAs, hsa_circ_0000190 and hsa_circ_0001649, secreted by LC cells in vitro, which were also detected in blood by RT-ddPCR. Further, they were able to monitor the expression of these two circRNAs in blood plasma of a cohort of LC patients, and correlated these circRNAs with a number of pathological parameters and response to immunotherapy. Authors conclude that circRNAs could be used as secreted biomarkers.
Overall, the manuscript is well-written. Methodology seems appropriate. Manuscript presents some interesting findings based on both in vitro and patient data.
Reply:
Thank you for your positive response on our manuscript.
However, there are some minor corrections required in the manuscript which are provided below.
1. Line 149, authors have cited Figure 1F. However, they have not provided this figure.
Reply:
Thank you for noticing this error. We have deleted this reference to Figure 1F, as it does not exist, and this sentence is just a summary for this Results section.
Authors are suggested to provide expression levels of hsa_circ_0000190 and hsa_circ_0001649 in lung cancer cell lines.
2. Line 164, Figure 1A should be given as Figure 2A.
Reply:
Corrected.
3. Authors should provide details of inclusion and exclusion criteria for patient population in the method section.
Answer:
Reply:
Thank you for this valid suggestions. We have added the following description to the Methods section: “Lung cancer patients who had received diagnosis, staging, and treatment were recruited into the present study. Patients were excluded if they had been diagnosed with other cancers; had incomplete medical records or had received less than 3 months of follow-up.”
4. Authors should provide details regarding cell line authentication and mycoplasma testing in cell culture method section.
Reply:
We have modified the Cell Culture section in the Materials and Methods to include the cell line authentication and mycoplasma testing: “Human lung cancer cell lines were obtained from the American Type Culture Collection (ATCC) before 2007 and tested positive for human origin. Cells were cultured in DMEM with 10 % fetal bovine serum, 100 units/mL penicillin, and 100 μg/mL streptomycin under standard culture conditions (37℃, 95 % humidified air, and 5 % CO2). Subculturing was performed using trypsin-EDTA. Medium was refreshed every two days. All cells lines were tested negative for mycoplasma contamination.”

Round 2
Reviewer 1 Report
The authors have added additional information on the bioinformatics, binding site information related to the circular RNAs. Also, they have made the raw data available in the GEO repositories. I agree with the others regarding the focus of the paper and scope for future studies. I recommend for the acceptance of the manuscript in the present form. It is an interesting study and can be of great value to the readers of the journal.